

# Genetic identification and hybridization in the seagrass genus *Halophila* (Hydrocharitaceae) in Sri Lankan waters

Shang Yin Vanson Liu[1,2,3], Terney Pradeep Kumara[4] and Chi-Hsuan Hsu[1]

[1] Department of Marine Biotechnology and Resources, National Sun Yat-Sen University, Kaohsiung, Taiwan
[2] Doctoral Degree Program in Marine Biotechnology, National Sun Yat-Sen University, Kaohsiung, Taiwan
[3] Graduate Institute of Natural Products College of Pharmacy, Kaohsiung Medical University, Kaohsiung, Taiwan
[4] Department of Oceanograhy and Marine Geology, University of Ruhuna, Matara, Sri Lanka

Corresponding author
Shang Yin Vanson Liu,
oceandiver6426@gmail.com,
syvliu@mail.nsysu.edu.tw

## ABSTRACT

Seagrasses, as marine angiosperms, play important roles in coastal ecosystems. With increasing anthropogenic impacts, they are facing dramatic declines on a global scale. *Halophila* is well-known as a complex taxonomic challenge mainly due to high morphological plasticity. By using only a morphological approach, the genus could be over-split or similar species could be erroneously lumped, thus masking its true biodiversity. In the present study, we incorporated genetic identification with morphological examination to reveal the identity of *Halophila* plants in southern and northwestern Sri Lankan waters. The nuclear ribosomal internal transcribed spacer (ITS) region and chloroplast ribulose-bisphosphate carboxylase gene (rbcL) were used to identify plants collected from the Gulf of Mannar, Puttalam Lagoon, and Matara, Sri Lanka. Based on genetic identification, *H. major* (Zoll.) Miquel is reported for the first time from Sri Lanka, which might have been misidentified as *H. ovalis* in previous literature based on morphology alone. We also observed a first hybridization case of *Halophila* cross between *H. ovalis* and *H. major*. Two potential cryptic species were found, herein designated *Halophila* sp. 1 (allied to *H. minor*) and *Halophila* sp. 2 (closely related to *H. decipiens*). In order to clarify taxonomic ambiguity caused by morphological plasticity and the low resolution of genetic markers, further comparative phylogenomic approaches might be needed to solve species boundary issues in this genus.

## INTRODUCTION

Seagrasses, a functional group of marine flowering plants found in coastal areas of the world's oceans, provide essential habitat for many coastal species and support marine food webs, playing critical roles in the balance of coastal ecosystems and human livelihoods (*Mtwana Nordlund et al., 2016*). It has been shown that seagrass habitat declined worldwide at a rate of 110 km$^2$ per year between 1980 and 2006 (*Waycott et al., 2009*). *Short et al. (2011)* suggested 72 seagrass species needed to be listed in the Red List of International Union for the Conservation of Nature (IUCN) based on global population status. Therefore,

there is an urgent need to conduct baseline studies (i.e., diversity, abundance, and distribution) for establishing conservation plans in the future. However, identification based on morphological traits in the genus *Halophila* is considered to be very challenging since few morphological differences or characteristics exist among closely related species (*Kuo et al., 2006*). Field ecologists without taxonomic knowledge of this genus may either overestimate or underestimate its true biodiversity (*Shimada et al., 2012*; *Tuntiprapas et al., 2015*; *Kurniawan et al., 2020*).

The genus *Halophila* comprises approximately 20 species within five sections based on morphological differences (*Den Hartog & Kuo, 2007*; *Kuo et al., 2006*). Most species in the genus are in section *Halophila*, which contains species with a pair of petiolate leaves borne on short, erect lateral shoots (*Den Hartog & Kuo, 2007*; *Kuo et al., 2006*). All other species are in sections *Microhalophila* (*H. beccarii*), *Spinulosae* (*H. spinulosa*), *Tricostata* (*H. tricostata*), and *Americanae* (*H. engelmannii* and *H. baillonis*). However, molecular genetic studies propose that *H. hawaiiana* and *H. johnsonii* should be treated as conspecific with *H. ovalis* (*Short, Moore & Peyton, 2010*). The ITS (internal transcribed spacer) sequence is proven to have great resolution for acting as a genetic barcode for the genus *Halophila* (*Kim et al., 2017*). *Kim et al. (2017)* showed that five major clades can be identified in section *Halophila,* which has relatively simple phyllotaxy compared to other sections. However, five morphologically similar species cannot be distinguished in the *Halophila ovalis* complex with ITS: *H. ovalis, H. minor, H. hawaiiana, H. johnsonii,* and *H. ovata*. On the other hand, that section's ITS region is capable of identifying *H. decipiens, H. major, H. nipponica, H. okinawensis, H. guidichudii,* and *H. stipulacea*. On the other hand, the rbcL (ribulose-bisphosphate carboxylase) gene of the chloroplast is suggested as a potential barcode region for land plants since it can discriminate among species in approximately 85% of congeneric pair-wise comparisons (*Newmaster, Fazekas & Ragupathy, 2006*). For seagrasses, the combined use of rbcL and matK (maturase K) genes is recommended by the Consortium for the Barcoding of Life (CBOL). However, neither the resolution of rbcL alone or the combination of rbcL and matK can well resolve the phylogenetic relationship of closely related species within the genus *Halophila* (*Nguyen et al., 2015*). Since *Halophila* species have notoriously great morphological plasticity, ITS and rbcL resolution incorporated with detailed morphological examination should provide valuable insight on the biodiversity of the genus in Sri Lanka.

Fourteen species belonging to six genera (60% of Indo-Pacific bioregion seagrasses) have been recorded in Sri Lanka: *Enhalus acoroides, H. beccarii, H. decipiens, H. ovalis, H. ovata, H. minor, H. stipulacea, Thalassia hemprichii, Cymodocea rotundata, C. serrulata, Halodule uninervis, H. pinifolia, Ruppia maritima,* and *Syringodium isoetifolium* (*Udagedara et al., 2017*). Among these, *H. ovalis, H. ovata,* and *H. minor* belong to the *H. ovalis* complex, which is very difficult to differentiate by morphological traits. *Udagedara et al. (2017)* have also mentioned that the distribution records of seagrasses in the Sri Lankan coast are extremely limited due to three decades of civil conflict ending in 2009 and a concurrent severe decline in seagrasses. A recent study showed that species composition changed and biodiversity decreased from 1991 to 2013 at Puttalam Lagoon in association with human

activities (*Ranahewa et al., 2018*). Therefore, there is an urgent need to understand the distribution and diversity of seagrasses in Sri Lankan waters before local extinctions occur.

In the present study, we incorporate morphological and genetic analyses on plants belonging to the genus *Halophila* collected from Sri Lankan waters that are difficult to identify by using either approach alone. With this integration, we attempt to reveal potentially overlooked biodiversity and species distribution.

## MATERIALS & METHODS

### Sampling and phylogenetic analyses

*Halophila* samples were collected by either snorkeling or sampling from the shoreline in early December 2018 under the permission No. NIC:196931003193 (issued by the Department of Wildlife Conservation) from three sites at depths from 0.5 m to 2 m based on occurrence data in previous literature and information from local fishermen: the Gulf of Mannar ($N = 32$; GPS: 8.9753739N, 79.9224197E), Puttalam Lagoon, Kalpitiya ($N = 32$; GPS: 8.3671739N, 79.785565E), and Matara ($N = 33$; GPS: 5.9474017N, 80.6349298E) (Fig. 1). The former two locations are within a lagoon system having very low visibility (<20 cm). *Halophila* plants collected from Puttalam were scattered below a huge meadow of *Thalassia hemprichii* that occurred along the lagoonal side of a sandbar. We collected *Halophila* plants only from the shoreline at Mannar, since the water is highly polluted by households. Conversely, the site in Matara where we collected *Halophila* plants faces a coastal ocean area dominated by a 10 x 50 m *H. major* meadow having >5 m visibility. Leaves were preserved in silica gel for further DNA extraction. Genomic DNA were extracted from leaves using a Plant Genomic DNA Mini Kit (Geneaid Biotech, Taipei, Taiwan). The two markers used in the present study were the rbcL gene from chloroplast DNA and nuclear ITS1-5.8S-ITS2. Primer pairs used in this study were P609 5′-GTAAAATCAAGTCCACCRCG-3′and P610 5′-ATGTCACCACAAACAGAGACTAAAGC-3′ for rbcL (*Lucas, Thangaradjou & Papenbrock, 2012*) and ITS5a 5′-CCTTATCATTTAGAGGAAGGAG-3′and ITS4 5′-TCCTCCGCTTATTGATATGC-3′ for ITS1-5.8S-ITS2 (*Nguyen et al., 2014*). Two loci were amplified in 25 μL reactions in a gradient thermocycler (Veriti 96-well thermal cycler, Thermo Fisher Scientific) over an initial denaturation step at 95 °C for 3 min, 30 cycles of denaturation at 94 °C for 30 s, annealing at 50 °C (ITS1-5.8S-ITS2) and 57 °C (rbcL) for 30 s, extension at 72 °C for 1 min, with a final extension step at 72 °C for 5 min. Each reaction contained 30 ng template DNA, 12.5 ul 2X master Mix RED (15 mM MgCl$_2$ and 0.4 mM each dNTPs),  200 nM of each primer, 0.2 unit of Ampliqon DNA polymerase (Ampliqon, Denmark) and dd water was added to make a final volume of 25 ul. PCR products were sent to Genomics (New Taipei City, Taiwan) for sequencing by an ABI 377 automated sequencer (Carlsbad CA, U.S.A.).

Known ITS sequences and rbcL from other *Halophila* species were added to the dataset for comparison (Table S1). Sequences obtained in the present study were aligned with reference sequences, including those within sections Spinulosae, Tricostata, Microhalophila, and Americanae as outgroups by MEGA 7 (*Kumar, Stecher & Tamura, 2017*) to visually inspect all alignments as well as search for the best nucleotide mutation model. Phylogenetic

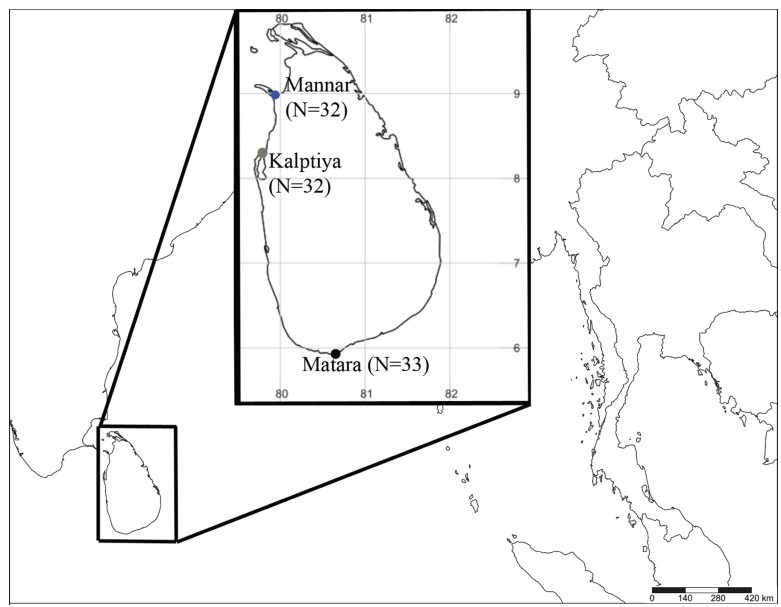

**Figure 1** Sample collection sites coded with different colors (blue: Mannar, green: Kalpitiya, and red: Matara) for *Halophila* surveys in Sri Lanka. Numbers in brackets indicate sample size.

analyses were performed to reveal genetic divergences among *Halophila* plants collected from different geographic locations, with Bayesian inference assessments through Mr Bayes (MB) version 3.2.2 (*Ronquist et al., 2012*) and maximum likelihood (ML) being performed by CIPRES Science Gateway (*Miller et al., 2015*). The former implemented two parallel runs of four simultaneous Markov chains for 10 million generations, sampling every 1000 generations and using default parameters. We discarded the first million generations (10%) as burn-in, based on the stationarity of log-likelihood tree scores. ML analyses were conducted in RAxML version 8.1.24 (*Stamatakis, 2014*) on CIPRES Science Gateway with default settings. Supporting value on the branches were evaluated by non-parametric bootstrapping with the automatically halt bootstrapping option by RAxML (ML).

## Molecular cloning

Among the three sites, a majority of the plants (22/32) collected from Puttalam Lagoon, Kalpitiya, failed to sequence on the ITS1-5.8S-ITS2 region due to multiple templates. We then obtained pure ITS1-5.8S-ITS2 sequences by using molecular cloning.

ITS1-5.8S-ITS2 PCR products amplified from SB21, SB22, SB23, and SB 24 were ligated into pJET1.2/blunt vector and cloned using a CloneJet PCR cloning kit (Thermo Scientific, U.S.A.). In total, 17 positive clones were selected for further PCR reaction. The final cloned PCR fragments were sequenced by Genomics (New Taipei City, Taiwan) using pJET1.2 forward and reverse primers. All sequences derived from the present study were submitted to GenBank under accession numbers MT347850–MT347937 (ITS) and MT422621–MT422718 (rbcL).

## Morphological analyses

One mature leaf was taken from 48 different plants in each category, comprising *H. major*, *H.* sp. 1 (allied to *H. minor*), *H. ovalis*, hybrids, potential hybrids, and *Halophila* sp. 2 (*H. decipiens* like) (Table 1), for morphological measurements consisting of lamina width, lamina length, distance from intramarginal vein to lamina margin, cross-vein angle, and number of cross-veins (Fig. S1). We calculated the ratio between intramarginal veins to the edge and the half-length of the width, and the ratio between lamina width and length. Specimens were identified using the keys of *Den Hartog & Kuo (2007)* and *Kuo et al. (2006)*. Morphological data were transformed (x-mean/standard deviation) and subjected to PCA to find out the variation among categories using Past3 software (*Hammer, Harper & Ryan, 2001*).

## RESULTS

### Phylogenetic analyses

For the ITS1-5.8S-ITS2 region, only 59 of 97 plants were successfully sequenced, 38 failing to achieve consensus sequencing due to the multiple template effect. We subsequently selected four samples for molecular cloning and obtained 17 sequences of the ITS1-5.8S-ITS2 region. In total, with 24 sequences obtained from GenBank and two sequences from *H. decipiens* collected from southern Taiwan, 14 valid species of *Halophila* were represented. In total 112 sequences were used for alignment and phylogenetic analyses. The length of aligned sequences is 615 bp, with 157 parsimony informative sites. Based on the ITS phylogenetic tree, sequences in the present study can be divided into three highly supported clades, which are *H. major*, the *H. ovalis* complex, and a potential new species (*Halophila* sp. 2) that is closely related to *H. decipiens* in terms of anatomic structure as seen under a SEM (J Kuo & S Liu, unpublished data). Interestingly, fresh plants of *Halophila* sp. 2 are very similar to *H. stipulacea* (Fig. S2). Outgroups are in sections Microhalophila (*H. beccarii*), Spinulosae (*H. spinulosa*), Tricostata (*H. tricostata*), and Americanae (*H. engelmannii*), which have complicated phyllotaxy and can clearly be separated from species having simple phyllotaxy in section Halophila, except *H. australis*. The basal clades of the ingroup comprised three species having leaf edge serrations, including *H. stipulacea*, *H. decipiens*, and *Halophila* sp. (*H. stipulacea* like clade). Sequences derived from plants collected from all three Sri Lankan sites fell into two main clades: the *H. ovalis* complex clade and *H. major* clade. Sequences clustering in the *H. major* clade were mainly from Mannar and Matara, whereas sequences clustering in the *H. ovalis* complex clade were from Mannar and Puttalam Lagoon. Interestingly, sequences derived from molecular cloning showed that any given plant contained ITS sequences of both *H. ovalis* and *H. major,* as shown in Fig. 2. For example, six sequences were obtained from SB24, two of them belonging to the *H. major* clade and the remainder to the *H. ovalis* complex clade. This may indicate a possible hybrid cross between *H. ovalis* and *H. major*.

Only one of the 97 samples failed to amplify rbcL. With eight sequences downloaded from GenBank representing seven species of *Halophila*, the length of the final alignment was 440 bp with only six parsimony informative sites. *H. beccarii* and *H. engelmannii* served

Liu et al. (2020), *PeerJ*, DOI 10.7717/peerj.10027

**Table 1  Comparisons of leaf morphology of *H. major*, *H. ovalis*, and *H. stipulacea* like species collected in Sri Lanka and previous studies.**

| Characteristic | Species | | | | | | | | | | |
|---|---|---|---|---|---|---|---|---|---|---|---|
| | **H. ovalis** | | | | | **H. major** | | | **H. stipulacea** | **H. decipiens.** | **Halophila sp2.** |
| | *Den Hartog (1970)* | Halophila sp. 1 (SB) | H. ovalis (SB) | Potential hybrid (SB) | Hybrid (SB) | *Kuo et al. (2006)* | H. major (MTR) | H. major (MA) | *Procaccini et al. (1999)* & *Vera et al. (2014)* | *Kuo et al. (2006)* | H. decipiens like (MA) |
| Number of samples | | 5 | 2 | 6 | 4 | | 13 | 10 | | | 8 |
| Lamina length (mm) | 10–40 | 4.42–6.61 | 11.00–16.00 | 12.00–20.00 | 23.80–33.00 | 15–25 | 19.80–31.00 | 8.05–31.13 | 63.8–84.3 | 20 | 17.75–40.36 |
| Lamina width (mm) | 5–20 | 1.15–1.89 | 3.90–6.89 | 6.55–9.90 | 8.60–14.13 | 9–11 | 10.50–13.75 | 4.72–14.63 | 6.5–8.43 | 4–6 | 2.75–3.89 |
| No. of pair cross veins | 10–25 | 3–4 | 13 | 11–17 | 18–27 | 14–17 | 11–17 | 12–18 | 11–18 | 6–9 | 8–15 |
| Space between intramarginal veins (mm) | 0.1–0.3 | 0.09–0.17 | 0.29–0.33 | 0.34–0.51 | 0.39–0.54 | 0.2 | 0.50–0.72 | 0.29–0.66 | 0.5 (*Ruiz & Ballantine, 2004*) | 0.25–3 | 0.24–0.39 |
| Cross-vein angles | 45–60° | 46–68° | 54° | 58–82° | 57–62° | 45–60° | 59–78° | 57–80° | 45–60° | n/a | 57–67° |
| LW:LL | n/a | 0.25–0.35 | 0.43–0.35 | 0.33–0.59 | 0.33–0.43 | n/a | 0.44–0.56 | 0.40–0.59 | n/a | 0.2–0.33 | 0.09–0.20 |
| HLW:DE | 0.5–0.63 | 4.95–10.03 | 10.51–6.75 | 7.90–12.94 | 10.29–13.25 | 0.71–0.33 | 9.01–12.69 | 8.27–6.32 | n/a | 2–3 | 4.89–6.36 |

**Notes.**

n/a,  not available.
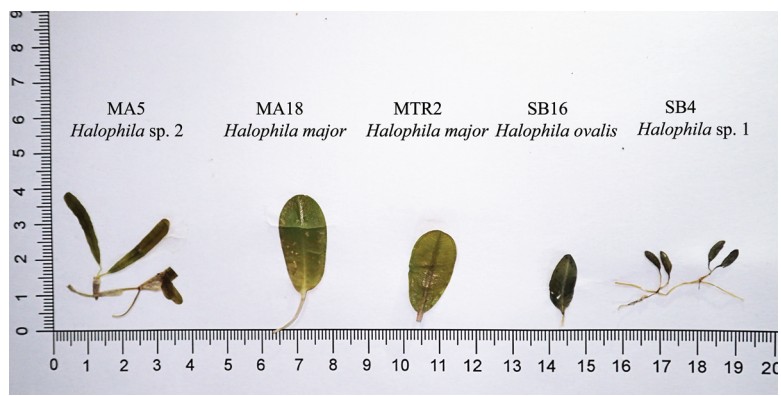

**Figure 2** Comparison of leaf morphology of *H. major*, *H. ovalis*, and *H. stipulacea* like specimens collected in Sri Lanka. Samples displayed in this figure were included and showed in ITS tree (Fig. 3).

as outgroups. At the basal-most position of the ingroup, *Halophila* sp. 2 clustered with a sequence derived from a sample collected from India that was identified as *H. stipulacea* and sister to *H. decipiens*. However, sequences derived from most Sri Lankan samples formed a monophyletic clade along with references identified as *H. ovalis*, *H. major*, and *H. minor* (Fig. 3). Since rbcL lacked genetic variation among the different *Halophila* species, the unresolved phylogeny indicates that rbcL cannot resolve species boundaries in *Halophila*.

## Morphological analyses

We were unable to measure all the plants that we collected in the present study because we did not collect enough shoots for both DNA extraction and morphological examination during sampling. After DNA extraction, we were able to measure 48 plants, which were further defined into six categories comprised of potential hybrid (failure to sequence without further cloning due to multiple templates), hybrid, *H. ovalis*, *Halophila* sp. 1 (allied to *H. minor*), *H. major,* and *Halophila* sp. 2 (*H. decipiens* like) (Fig. 4). The number of plants examined and results of five measurements and two ratios are given in Table 1. Raw data were transformed and a principal component analysis (PCA) was performed. Variance explained by the first two PCA components (PCA1 and PCA2) is 79.576%. The majority of the variance (95%) of PCA-1 and PCA-2 was explained by lamina length and lamina width:lamina length, respectively. The result of the PCA plot shows that *Halophila* sp. 1 (allied to *H. minor*) and *Halophila* sp. 2 can be distinguished from other categories by smaller lamina length and smaller lamina width:lamina length, respectively. *H. major* had the widest distribution in the PCA plot, and the hybrid and potential hybrid fell within the range of *H. ovalis* and *H. major* (Fig. 5).

## DISCUSSION

The literature related to seagrass communities and biodiversity in the Sri Lanka is scarce. *Jayasuriya (1991)* mentioned that there are 12 species among nine genera recorded from Sri Lanka, the number being increased to 15 in 2007 (*De Silva & Amarasinghe, 2007*). The latest report showed 14 species as of 2017 (*Udagedara et al., 2017*), with the complete list

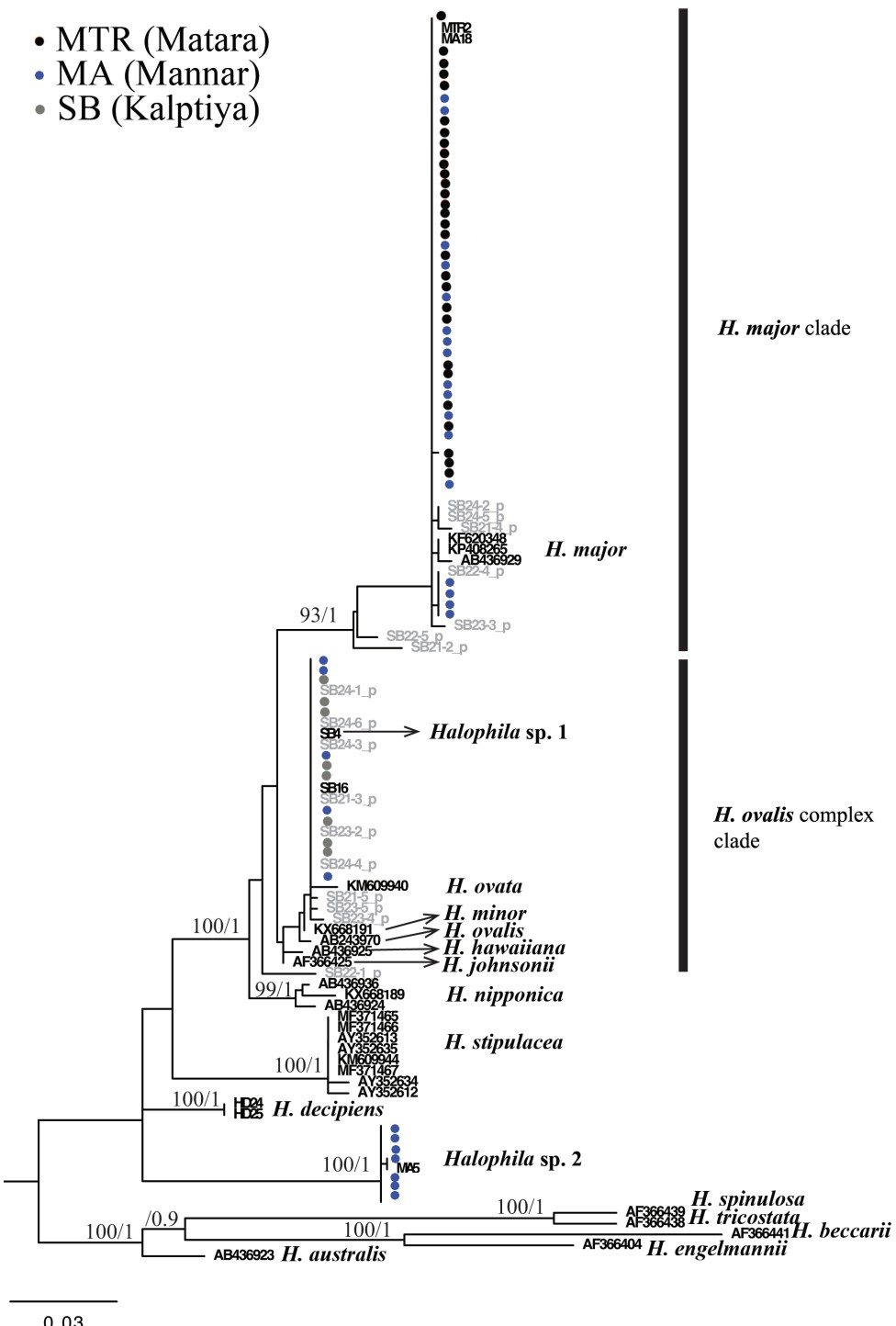

**Figure 3** **Phylogeny of *Halophila* inferred from maximum likelihood and Bayesian analysis based on 615 bp (including gaps) of nrDNA sequences comprising ITS-1, 5.8S rDNA and ITS-2.** Nodes are presented only for those with bootstrap scores > 90% majority rule for maximum likelihood and > 90% majority probabilities for Bayesian probability values (ML/BI). Sequences are color-coded based on different sampling locations as in Fig. 1. Sample names with shading are sequences derived from molecular cloning.

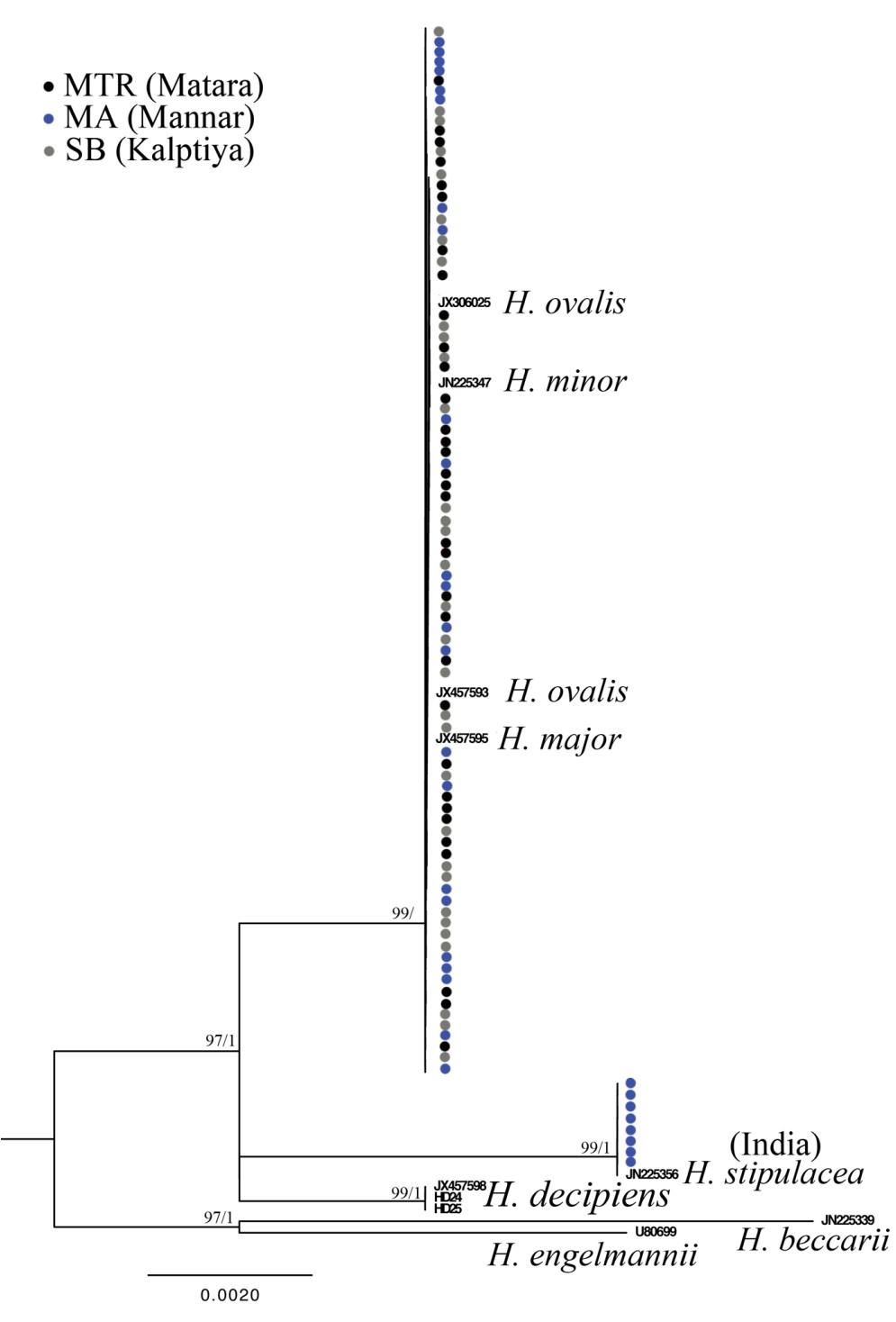

**Figure 4 Phylogeny of *Halophila* inferred from maximum likelihood and Bayesian analysis based on 440 bp of the rbcL gene.** Nodes are presented only for those with bootstrap scores > 90% majority rule for maximum likelihood and > 90% majority probabilities for Bayesian probability values (ML/BI). Sequences are color-coded based on different sampling locations as in Fig. 1.

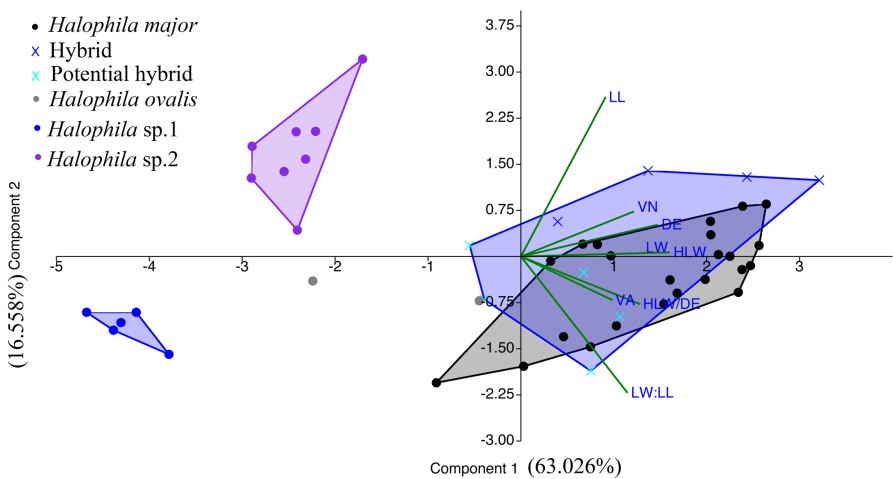

**Figure 5** **Component loadings for the first of two principal components of the PCA of morphological traits with convex hull of different sample groups.** LL, lamina length; LW, lamina width; VA, crossed vein angle; DE, distance between marginal vein and lamina edge; LW/LL, lamina width/lamina length, and HLW/DE, half lamina width/distance between marginal vein and lamina edge.

being *Enhalus acoroides*, *Halophila beccarii*, *H. decipiens*, *H. ovalis*, *H. ovata*, *H. minor*, *H. stipulacea*, *Thalassia hemprichii*, *Cymodocea rotundata*, *C. serrulata*, *Halodule uninervis*, *H. pinifolia*, *Ruppia maritima*, and *Syringodium isoetifolium*. In the present study, *H. major* (Zollinger) Miquel (1855) is a new species record for Sri Lanka based on genetic analyses. *H. major* was previously treated as a synonym of *H. ovalis* by *Den Hartog (1970)*, and in 2006 *Kuo et al. (2006)* examined global type materials and concluded that reinstating the taxon status of *H. major* was warranted. Additionally, further phylogenetic studies (*Uchimura et al., 2008*; *Nguyen, Holzmeyer & Papenbrock, 2013*) showed molecular evidence that *H. major* can also be separated from *H. ovalis* by using the ITS region. *Kuo et al. (2006)* suggested that *H. major* and *H. ovalis* can be distinguished by a ratio of 1/2 of the lamina width to the distance between the intramarginal veins and lamina margin, and the number of cross-veins. However, most of our measurements of *H. ovalis* and *H. major* overlap (Table 1). This morphological plasticity could lead to underestimating seagrass biodiversity as mentioned in *Nguyen et al. (2014)*. In the recent survey of the Gulf of Mannar, Puttalum Lagoon, and southern Sri Lanka (*Ranahewa et al., 2018*; *Ranatunga & Pethiyagoda, 2015*; *Gunasekara, 2017*), *H. major* may mistakenly be identified as *H. ovalis*. In addition, our phylogenetic analysis also points out that *H. major* could be widely distributed in Sri Lanka, since plants collected from Mannar and Matara are identified as *H. major*. Potential hybrids from Puttalum Lagoon have both *H. ovalis* and *H. major* ITS sequences, which may indicate the presence of *H. major* actually occurring where we failed to collect it. The ITS phylogenetic tree also showed that the *H. decipiens* like plants (*Halophila* sp. 2) collected from the Gulf of Mannar are distinct from *H. decipiens* in Taiwan. Conversely, rbcL sequences derived from *H. decipiens* like plants cluster with *H. stipulacea* collected from India. This incongruence could possibly result in erroneous identifications since

*Halophila* sp. 2 resembles *H. stipulacea* in the field. Another possible explanation is the lack of genetic variation on the rbcL chloroplast gene, which failed to resolve species boundaries in the genus *Halophila* (*Lucas, Thangaradjou & Papenbrock, 2012*). The present study also shows that there are only six parsimony informative sites across seven *Halophila* species among 440 bp.

PCA analyses based on morphology show that *Halophila* sp.1 (allied to *H. minor*) and *Halophila* sp. 2 (*H. decipiens* like) can be separated from other categories. Although *Halophila* sp. 2 is very similar to *H. stipulacea* in appearance, most measurements are smaller than plants in the Mediterranean Sea (*Procaccini et al., 1999*) as well as those described as *H. decipiens* in *Kuo et al. (2006)*. As mentioned by *Den Hartog (1970)*, Indian Ocean *H. stipulacea* plants often have delicate and membranous but never bullate leaves, and more or less deciduous stipules. These plants were initially collected by Isaac Bailey Balfour during the Transit of Venus expedition at Rodrigues Island in 1874 (*Balfour, 1879*), and later described as *Halophila balfourii* Soler (*Solereder, 1913*). Currently, it is treated as a synonym of *Halophila stipulacea* (Forsskål) Ascherson. Therefore, the plants collected from the Gulf of Mannar could be *H. balfourii*. However, further genetic analyses from a broad sampling across its current distribution, including the population from the type locality of *H. stipulacea* and *H. decipiens* in the Red Sea, is needed to clarify the identity of *Halophila* sp 2. Most *Halophila* sp.1 (allied to *H. minor*) measurements overlap with *H. minor* or *H. ovate* (*Kuo & Den Hartog, 2001*), except that lamina width (1.15–1.89 mm) is smaller compared to these two species (*H. minor*: 3.5–6 mm; *H. ovata*: 4-8 mm). Unfortunately, even by using combination loci including matK, rbcL, and trnH-psbA (*Lucas, Thangaradjou & Papenbrock, 2012*) or ITS, rbcL, and matK (*Nguyen et al., 2015*), there was a failure to resolve species boundaries in the *H. ovalis* complex. Further comparative phylogenomic approaches (*Liu et al., 2017*; *Yu et al., 2018*) may be useful in resolving *H. ovalis* complex species boundaries.

*Soltis & Soltis (2009)* suggested that natural hybridization could be an important creative force and evolutionary process responsible for the increasing of angiosperm species diversity. The incongruence between phylogenetic relationships constructed based on different markers can be considered a signature of hybridization, as well as two divergent alleles of a single locus found in one individual. Intra-species variation in ITS have been identified in many different plant groups, which may hamper attempts to uncover accurate phylogenetic species relationships (*Poczai & Hyvönen, 2010*). Meanwhile, the high intraspecific variation in ITS is considered as incomplete concerted evolution driven by hybridization (*Xu et al., 2017*). Additionally, the maternal inheritance of the chloroplast gene tree (i.e., rbcL tree in the present study) reflects only the evolutionary processes of maternal lineages, which may mask genetic evidence of hybridization (*Okuyama et al., 2005*; *Soltis & Soltis, 2009*). Either of these reasons may cause incongruence between ITS and plastid phylogenies. Among marine angiosperms, natural hybridization has been observed in only four genera (*Halodule, Ruppia, Posidonia,* and *Zostera*) (*Ito & Tanaka, 2011*; *Coyer et al., 2008*; *Martínez-Garrido et al., 2016*; *Sinclair, Cambridge & Kendrick, 2019*). *Ito & Tanaka (2011)* found sympatric *Halodule uninervis* and *H. pinifolia* hybridizing in the waters of Okinawa by reconstructing their phylogenetic relationship with rbcL and

psbA-trnH loci. The congruent pattern between morphological traits and nuclear loci was also observed in two sympatric species of *Posidonia* (*P. australis* and *P. coriacea*) in Australia that show signs of hybridization (*Sinclair, Cambridge & Kendrick, 2019*). In the present study, the majority of *Halophila* samples collected from Kapitya failed to sequence due to multiple templates found in single plants, but 17 pure sequences in the ITS region were obtained with further cloning. Phylogenetic analyses showed that a single plant contained ITS sequences clustered with both *H. ovalis* and *H. major* (Fig. 2). The percentage of sequencing failure due to multiple templates varied among the three sites (22/32 at Kalpitiya, 1/35 at Mannar, and 5/33 at Matara), indicating that hybridization may be common at these three sites, especially Kalpitiya. However, the PCA plot based on morphology showed that most traits overlap among hybrid, *H. ovalis,* and *H. major.* This may be due to the morphological plasticity found in *Halophila* (*Den Hartog & Kuo, 2007*; *Kuo et al., 2006*; *Singh, Southgate & Lal, 2019*).

## CONCLUSIONS

In conclusion, *Halophila* plants collected from Sri Lanka cluster into three clades by the ITS tree, represented as *H. major*, *H. ovalis* complex, and *Halophila* sp. 2 clade. *H. major* is recorded as a new species of the genus *Halophila* in Sri Lanka, and may have a wide distribution and possibly be misidentified as *H. ovalis* in the previous literature. Meanwhile, *H. decipiens* like plants collected from Mannar may represent a cryptic species of either *H. stipulacea* or *H. decipiens* based on phylogenetic relationship traits shared among them. Surprisingly, we found the first case of hybridization in the genus *Halophila,* which may be a cross between *H. ovalis* and *H. major.* Further phylogeographic study with a broader sampling scheme that includes plants from type localities and applying methods based on massive parallel sequencing (i.e., Hyb-Seq, review in *Yu et al., 2018*) that can obtain genome wide genetic variation is needed to clarify the taxonomic status of *Halophila* sp. 1 and sp. 2.

## ACKNOWLEDGEMENTS

Special thanks to Dr. John Kuo for his feedback on the preliminary draft and Mr. MAK Peiris for assistance during the 2018 field trip.

### Funding

This work was supported by the Ministry of Science and Technology of Taiwan (MOST) (No. 107-2911-I-110-301-, 108-2911-I-110-301, 107-2611-M-110-015- and 107-2119-M-110-009-). The funders had no role in study design, data collection and analysis, decision to publish, or preparation of the manuscript.

### Grant Disclosures

The following grant information was disclosed by the authors:

Ministry of Science and Technology of Taiwan (MOST): 107-2911-I-110-301-, 108-2911-I-110-301, 107-2611-M-110-015-, 107-2119-M-110-009-.

## Competing Interests

The authors declare there are no competing interests.

## Author Contributions

- Shang Yin Vanson Liu conceived and designed the experiments, performed the experiments, analyzed the data, prepared figures and/or tables, authored or reviewed drafts of the paper, and approved the final draft.
- Terney Pradeep Kumara performed the experiments, authored or reviewed drafts of the paper, and approved the final draft.
- Chi-Hsuan Hsu performed the experiments, analyzed the data, authored or reviewed drafts of the paper, and approved the final draft.

## Field Study Permissions

The following information was supplied relating to field study approvals (i.e., approving body and any reference numbers):

Department of Wildlife Conservation issued the permit to collect seagrass samples from Sri Lankan waters (NIC:196931003193).

## Data Availability

Data is available at GenBank: MT347850–MT347937 (ITS) and MT422621–MT422718 (rbcL).

## Supplemental Information

Supplemental information for this article can be found online at http://dx.doi.org/10.7717/peerj.10027#supplemental-information.

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
