# Peer review of "Genetic identification and hybridization in the seagrass genus Halophila (Hydrocharitaceae) in Sri Lankan waters"

_PeerJ, doi:10.7717/peerj.10027_

## Round 0.1 · original submission · Minor Revisions

I first wanted to apologize for the unusually long time it took to review your manuscript - I hope that the delay has not been a hardship to you. We now have feedback from three referees who agree that your paper is of value and ought to be published following a rather extensive list of suggested revisions.

As you will see, each referee has a somewhat different focus and set of suggested revisions. In sum, the comments are extensive, and I debated about whether the suggested revisions constitute a major or minor revision of the submitted manuscript. Ultimately, although the feedback from the referees is long and detailed, I see nothing that involves generation of additional data or new analyses that would merit a major revision and additional round of review.

As such, I believe that this is a rather extensive, but minor, revision that can be accomplished by a careful edit of the submitted text. I expect that if you complete a thorough edit of the manuscript that includes all the suggested improvements suggested by the referees, this manuscript is likely to become acceptable for publication.

When you revise your manuscript in response to this feedback, please also include a detailed point-by-point response to each referee so that I can see your response to the referee comments and exactly how the manuscript has been changed in response. I look forward to seeing your revised manuscript.

·

Basic reporting

I have read and reviewed the article entitled 'Genetic identification and hybridization in the seagrass genus Halophila (Hydrocharitaceae) in Sri Lankan waters' and overall find the manuscript well written in a professional English tone with only a few punctuation and spelling errors (line 29 - period instead of common after Kurniawan et al.; line 55 - space after 1991, Lines 83-85 & elsewhere - consistency in spacing between value and uL; . Line 198 and 207 - Use or no use of hyphen with 'H. decipiens-like'; Line 259 - misspelling of H. stipulacea; etc.)

The referenced literature is largely sufficient to frame the study, although it is unclear where all referenced measurements from Vera et al. (2014) are sourced from. For example, I don't believe in Vera et al. (2014) the blade width reported but appears in Table 1 here, unless measurements were approximated from the herbarium press photograph. The authors may want to consider other morphological measurements for H. stipulacea from the Caribbean for which several are available before and after Vera et al. 2014. If not included in the table, they could be discussed after Line 207-210.

The structure, figures, and tables are professionally prepared. One suggestion is to remove the decimal place on the numbered axes of PCA figure as they seems unnecessary and somewhat distracting. The scale used on the PCA also seems to bunch up values along the X-axis. I suggest re-scaling the Y-axis to -/+ 100,000 only and then stretching the figure to be taller.

Also, please include latitude/longitude grid/markings on Figure 1 (map of Sri Lanka) for readers unfamiliar with this geographic region. A regional map/insert would be helpful as well. What is the significance of the coding in Figure 2 - MA5, MA18? Kindly explain in legend. I find the in-tree sample labels on Figure 3 and Figure 4 very hard to read - the font size is very small. Can this be improved upon? Perhaps color coded by sampling location/source?

The paper is self-contained with relevant results presented and discussed.

Experimental design

The research is original and fits with the aims and scope of PeerJ. Furthermore, it is quite timely given increasing interest in this seagrass genus across its members' native and non-native ranges as outlined in recent review papers.

The research question is clear, but no explicit hypothesis is stated. Perhaps one could be crafted and included in lines 58-60. Inclusion of statistical hypotheses may be appropriate for strengthening the case of potentially novel/cryptic/hybrid Halophila species/plants found in Sri Lanka. These could be framed around the degree of genetic or morphological differentiation among species and tested via appropriate inferential statistics. Not critical for the paper, but why were morphological data transformed (lines 120-122; lines 166-167) for the PCA? What was the justification to transform instead of using straight values?

Most methods used are well-vetted and described with sufficient detail (the exception would be the cloning/testing for hybrids which I lack expertise to judge).

Validity of the findings

Findings seem valid and well-supported. The contribution of new knowledge on Halophila seagrasses in Sri Lankan waters is also very much appreciated.

One additional note - I am not familiar with the angiosperm literature on hybridization and what metrics are typically used to justify plant hybrids in the same genus, but it appears to me the authors have sufficiently explained this in the discussion. I defer to other reviewers’ more knowledgeable on the topic. As they state in lines 260-263, I encourage the authors to investigate this novel discovery.

Additional comments

No comment.

Reviewer 2 ·

Basic reporting

The ms is written well, and you give a great background and wrote a nice introduction. It is great that you give good background about the resolution of ITS. Please do the same for rbcL. As many readers my have no knowledge on details of molecular identification, consider to also add information on what these markers are (i.e. internal transcribed region, which…). Please also provide some details, on why you chose the 3 assessed sites and if they are representative of much of the surrounding area.

References are appropriate.

The first 3 figures are great. The phylogenies are so big that they are difficult to read. Please consider how you can improve them. Can you eg use symbols instead of individual names? Not all writing on the PCA will be readable. Do you need all ticks with writing? Do you need to report eg 5000.0? Increase font size of writing of component 1 and 2. Table 1 was cut off for me, so can’t comment on entire table. The summary states that there are 6 figures and 2 tables, but I can only see 5 figures and 1 table.

The sequences derived from cloning have been submitted to GenBank and accession numbers are given (I have not checked if they are accessible), others are given in supplementary. I thank you for providing the raw data, however your supplemental files need to be referenced to in the main ms and needsmore descriptive metadata identifiers.

Experimental design

The research question is well defined & meaningful. You make a very good case about the relevance of this work to record biodiversity and distribution of species in Sri Lanka.

The methods are appropriate, especially for an initial study that may lead the way to phylogenomic studies. The parameters collected are also valuable, but I am curious also about flowering rates. Did you observe flowers at any site?

Validity of the findings

The conclusions are well stated and confirmed by the data. Make sure you repeat in your conclusion the content of the following sentence from the abstract, as that might be of considerable interest for some: H. major (Zoll.) Miquel is reported for the first time from Sri Lanka

Additional comments

Abstract L15: Taxonomic
Intro
L23: Do you mean ALL seagrass spp?
L38: Needs ref here
M&M
L79: Please give refs for primers
Results
L157: It DOES in fact indicate that, yes.

·

Basic reporting

The literature related to seagrass communities and biodiversity in Sri Lanka is very scarce. Here authors apply molecular markers - bar coding markers - to demonstrate that the species diversity is actually larger than previously presumed, especially in the Halophila genus.

Overall, the manuscript is well written and, following what I consider a major revision, should be accepted.
I would ask that the authors refer to Les et al. 1997, 2006 and Procaccini et al. 1999 (Structural, morphological and genetic variability in Halophila stipulacea (Hydrocharitaceae) populations in the western Mediterranean) – all of which are super relevant to the study here.
The authors could have wrapped up the paper with something relating the results to future biodiversity conservation efforts in the region…this is important for explaining the broader impacts of this work.
Lastly, the authors totally neglect to discuss the relevance of their methods in the new age of next-generation sequencing (NGS) which could be of particular relevance to DNA barcoding of genetically close species (e.g., DOI: 10.1111/1755-0998.12236, DOI: 10.1007/978-1-61779-609-8_2). In this respect, the methods used in this paper are, by now, “ancient” and in my eyes are history and should not be encouraged…


Clear, unambiguous, professional English language was used throughout this manuscript.
Intro & background are written well.
Authors do neglect some basic papers that are very closely related to their study - examples including papers by Les et al. 1997, 2006 and Procaccini et al. 1999.
Literature referenced can be greatly improved – I have mentioned where…many issues.
Books are not referenced properly…revise the formatting. Missing important papers
article structure, figures, tables - are in order.

Experimental design

The exp design is ok - but raw sequences have not been uploaded in public banks (e.g., NCBI).

Validity of the findings

Building on from the latest report that showed that Sri Lanka had 14 seagrass species as of 2017 (Udagedara et al., 2017), in the present study, based on genetic analyses, the authors show that some plants previously considered H. ovalis based only on morphological differences), are actually H. major (Zollinger) Miquel (1855), which is a new species record for Sri Lanka (thus, bringing the species list to 15). These results support the notion that relying only on morphological measurements, could lead to underestimating seagrass biodiversity due to morphological plasticity.
Authors also point out to the existence of potential hybrids that have both H.ovalis and H. major ITS sequences. On the other hand, authors show that the rbcL chloroplast gene failed to resolve species boundaries in the genus Halophila.
This study demonstrates again the complexity of the H. ovalis species boundaries in particular, and the morphological plasticity found in Halophila in general.


Overall, the manuscript is well written and following revision should be accepted.
I would ask that the authors refer to Les et al. 1997, 2006 and Procaccini et al. 1999 (Structural, morphological and genetic variability in Halophila stipulacea (Hydrocharitaceae) populations in the western Mediterranean) – all of which are super relevant.
The authors could have wrapped up the paper with something relating the results to future biodiversity conservation efforts in the region…
Lastly, the authors totally neglect to discuss the relevance of their methods in the new age of next-generation sequencing (NGS) which could be of particular relevance to DNA barcoding of genetically close species (e.g., DOI: 10.1111/1755-0998.12236, DOI: 10.1007/978-1-61779-609-8_2).

Additional comments

Abstract:
L 6-7: “…we incorporate genetic identification with morphological examination to reveal the identity of Halophila plants in Sri Lankan waters.”
Since this work was already done, use past simple throughout the text - incorporated
Sri Lankan waters – all around the island, or in one specific region ? why not “…in southern and northwestern Sri Lankan waters”

L 16-17: “…the low resolution of genetic markers, further comparative phylogenomic approach might be needed to solve species boundary issues in the genus Halophila”
Here the authors already admit one of the major flaws of their markers that in these days are considered very old, with very low resolution.

Introduction:
L 20: “habitat” – maybe “habitats”.
L 19-21 – this is a very simplified and limited description on the services and functions of seagrasses. Also, in my eyes there are much better and newer citations that could be used here in addition to Short et al. 2011..nothing was even mention about their estimated value…you don’t need to repeat things that we’ve read many times, but at least cite other papers….

L 21: “has been shown that seagrass habitat disappeared worldwide at a rate of 110 km2 per year…”
Change “disappeared” to “declined”…
L 23: Short et al. (2011) suggested 72 species of – suggested THAT….
If you use Short here, use 2-3 different ones above so as not to use the same one….more relevant here.
L 24-25: red list of International Union for the Conservation of Nature (IUCN) – change to “Red List of ….”. before you go on explain what is the Red List Category Criteria for at least the genus of Halophila, “while most species of Halophila are not in danger, and seem to be proliferating, there are some species within this genus that listed as very vulnerable….

L 27: change “or unclear characters” to “characteristics”
L 30-31: there is no Den Hartog et al. 2006 in the reference list. There is a “Den Hartog, C., & Kuo, J. (2007)” which means it is not 2006 and it is not “et al.”
Please get to know Les et al. 1997, 2006.
Are you sure you want to go with “sections” ? why not “clades” ?
Why not introduce a figure showing the different morphologies of the different family types ?

L 51: In addition, Udagedare et al. (2017) have…
How is this “in addition” ? this is a totally new paragraph/topic…

Methods:
L 64 – so add info explaining that these are shallow or even intertidal plants
Add GPS info for each location
L 116-118 : “…for morphological measurements consisting of lamina width, lamina length,
117 distance from intramarginal vein to lamina margin, cross-vein angle, and number of cross-veins…” – so why not add another figure/insert showing an example of where does one find these parts on the leaf?
This will allow others to follow…

Results:
L 138-139: “…which have complicated phyllotaxy and can clearly separated from species having simple phyllotaxy in section Halophila, except H. australis”.
This seems to fit the discussion more than the results section – either way, I would suggest adding a reference to support this conclusion.

L 156-158: Since rbcL lacked genetic variation among the different Halophila species, the unresolved phylogeny may indicate that rbcL cannot resolve species boundaries in Halophila.
This seems to fit the discussion more than the results section – either way, I would suggest adding a reference to support this conclusion – have other studies, in other organisms, pointed out to the low resolution of this genetic marker ?
L 168 - change “is” to “was”.

Discussion:
L 191-4: This morphological plasticity could lead to underestimating seagrass
biodiversity in our recent survey of the Gulf of Mannar, Puttalum Lagoon, and southern Sri
Lanka (Ranahewa et al., 2018; Ranatunga & Pethiyagoda, 2015; Gunasekara, 2017) by
mistakenly identifying H. major as H. ovalis.

This is not unique to seagrasses. Bring examples of other cases where ITS-2 had a stronger resolution than morphology – see zooxanthellae clade diversity for example in papers by Todd C. LaJeunesse. Same story

L 203-204: “…rbcL chloroplast gene, which failed to resolve species boundaries in the genus Halophila…” – bring other examples of lower resolution of this marker gene – you are not the first case…


References:

Please go over one by one – many mistakes, many different formats for example:

Den Hartog, C., & Kuo, J. (2007). Taxonomy and biogeography of seagrasses. In Seagrasses:
biology, ecology and conservation (pp. 1-23). Springer, Dordrecht
- who are the editors? how many pages in book ?
-
L 290-292: Kuo, J., KANAMOTO, Z., IIZUMI, H., & MUKAI, H. (2006). Seagrasses of the genus Halophila Thouars (Hydrocharitaceae) from Japan. Acta Phytotaxonomica et Geobotanica, 57(2),129-154
Why the capitol names ??? Use same format for all papers.

336-337 Short, F. T., Moore, G. E., & Peyton, K. A. (2010). Halophila ovalis in the tropical Atlantic Ocean. Aquatic Botany, 93(3), 141-146.
All latin names s

Figures:
Fig 1 – missing arrow north, add coordinates on left and bottom sides
Fig 2 – so why not add arrows to the morphological you used for the measurements (at least on one species ) ?
Legend of Fig 3 seems to be double/confusing / something wrong there
Not clear which sequences were the out group…

---

## Round 0.2 · accepted · Accept

I have read your rebuttal letter and your revised manuscript and am satisfied with your responses. In the few cases where you have not made the changes suggested by the referees, you clearly explain why and how you dealt with the comment in another way. I find that I am in agreement with your responses, am satisfied with your revisions of the original submission, and cannot see anything of substance raised by referees that has not been changed as requested. As such, I see no reason to waste the time of the reviewers or further delay acceptance of your manuscript and am happy to move it forward into production.